# Synthesis, Properties and Photocatalytic Activity of CaTiO_3_-Based Ceramics Doped with Lanthanum

**DOI:** 10.3390/nano12132241

**Published:** 2022-06-29

**Authors:** Maxim V. Zdorovets, Daryn B. Borgekov, Inesh Z. Zhumatayeva, Inesh E. Kenzhina, Artem L. Kozlovskiy

**Affiliations:** 1Laboratory of Solid State Physics, The Institute of Nuclear Physics, Ibragimov St., Almaty 050032, Kazakhstan; mzdorovets@gmail.com (M.V.Z.); bogrekov@mail.ru (D.B.B.); kenzhina@physics.kz (I.E.K.); 2Engineering Profile Laboratory, L.N Gumilyov Eurasian National University, Satpayev St., Nur-Sultan 010008, Kazakhstan; inesh.zhumatayeva@gmail.com; 3Department of Intelligent Information Technologies, Ural Federal University, 620075 Yekaterinburg, Russia; 4Department of General Physics, Satbayev University, Almaty 050032, Kazakhstan

**Keywords:** doping, calcium titanate, photo-catalysis, decomposition of organic dyes, phase transformations

## Abstract

The aim of this work is to study the effect of lanthanum doping on the phase formation processes in ceramics based on CaTiO_3_, as well as to evaluate the effectiveness of the ceramics as photocatalysts for the decomposition of the organic dye Rhodamine B. The methods used were scanning electron microscopy to evaluate the morphological features of the synthesized ceramics, X-ray diffraction to determine the phase composition and structural parameters, and UV-Vis spectroscopy to determine the optical properties of the ceramics. During the experiments it was found that an increase in the lanthanum dopant concentration from 0.05 to 0.25 mol leads to the formation of the orthorhombic phase La_0.3_Ca_0.7_TiO_3_ and the displacement from the ceramic structure of the impurity phase TiO_2,_ which presence is typical for the synthesized ceramics by solid-phase synthesis. On the basis of the data of the X-ray phase analysis the dynamics of phase transformations depending on concentration of lanthanum was established: CaTiO_3_/TiO_2_ → CaTiO_3_/La_2_TiO_5_ → CaTiO_3_/La_0.3_Ca_0.7_TiO_3_ → La_0.3_Ca_0.7_TiO_3_. During the determination of photocatalytic activity it was found that the formation of La_0.3_Ca_0.7_TiO_3_ phase leads to an increase in the decomposition rate as well as the degree of mineralization.

## 1. Introduction

The annual increase in production related to the application of color coatings, coloring of materials in order to provide them with bright colors, and printing is accompanied by an increase in the use of various dyes, including organic dyes [1,2,3]. Moreover, the use of dyes in the textile and printing industry increases the amount of harmful emissions into wastewater and increases the level of its pollution. In most cases, organic dyes such as indigo carmine, cargo red, and Rhodamine B allow materials to be given bright, saturated colors, which enhance their widespread use [4,5,6]. However, due to their nature, these dyes are not amenable to disposal and neutralization when released into wastewater by classical methods such as mechanical filtration or capture, chemical absorption, etc. Such difficult disposal of these dyes is due to their high resistance to external influences, as well as chemical attack. In turn, their accumulation in the aqueous environments of wastewater can have a negative impact on the environment, as well as cause mutational effects on flora and fauna with prolonged interaction and large concentrations of accumulated dyes. All this requires new solutions in the field of disposal and neutralization of waste organic dyes, as well as their neutralization and decomposition into harmless components [7,8,9,10].

One of the most promising ways to utilize and neutralize organic dyes is their decomposition using catalysts when exposed to ultraviolet light. The prospects of photocatalytic decomposition reactions have been confirmed by a number of experimental works, which show that the catalysts themselves, used for photocatalytic decomposition and mineralization reactions, play a role in most cases of photocatalytic decomposition. A number of requirements are imposed on the materials of photocatalysts, which determine the range of their applications [11,12,13].

One of the key requirements for photocatalysts is their photoactivity due to the optical and electronic properties of the material, the band gap width and the rate of generation of photoelectrons under the influence of ultraviolet [14,15]. Another important requirement to photocatalysts is their stability over a long time of operation, and preservation of efficiency at long stay in various aqueous or aggressive environments. These properties allow the material to remain stable for a sufficiently long time not only in photocatalytic reactions, but also during operation for long time intervals [16,17].

One of the promising materials in this direction, in particular, photocatalytic decomposition of organic dyes, are nanostructured ceramics based on titanates, which have proven their effectiveness in photocatalysis due to the combination of their optical, structural, and segment-electric properties [18,19,20]. However, despite the sufficient number of scientific research studies in this direction and the large amount of known literature data in the field of photocatalytic decomposition, interest to this direction has not stopped. In the last few years, much attention has been paid to studying the possibilities and prospects in the direction of increasing not only the efficiency of photocatalytic reactions, but also increasing their resistance to external influences and corrosion processes [21,22,23,24,25]. In this regard, one of the avenues in this direction of research is the method of doping titanates with rare-earth elements, which allows not only increasing of their resistance to external influences, but also changing their electronic and optical properties. As shown in a number of works [26,27,28], doping with lanthanum makes it possible to introduce significant changes in the electronic and optical properties of perovskite-like ceramics, as well as to significantly increase their efficiency as photocatalysts or solar cells. For example, it was shown in [27] that doping of strontium titanium with lanthanum makes it possible to obtain oxygen-deficient La_0.5_Sr_0.2_TiO_2.95_ ceramics comparable in properties to zirconium dioxide. It was shown in [28] that doping with lanthanum leads to an increase in conductivity and catalytic activity.

Based on the above, the main purpose of this study is to investigate the effect of doping CaTiO_3_ ceramics with lanthanum at different concentrations to increase not only photocatalytic activity, but also to study the influence of the phase composition of ceramics on the optical and strength properties. Doping with lanthanum is caused by the possibility of increasing not only the resistance of perovskite-like ceramics to corrosion and mechanical damage, but also increasing photo-catalytic activity [25,26,27,28].

## 2. Experimental Part

Synthesis of perovskite-like ceramics CaTiO_3_ doped with lanthanum (La) was carried out in two stages. The first stage consisted in suspension of initial salts CaCO_3_, TiO_2_ (anatase), La(NO_3_)_3_ in the set stoichiometric parity: CaCO_3_:TiO_2_ → 0.5:0.5. La doping was carried out by addition of La(NO_3_)_3_ in quantity 0.05, 0.10, 0.15, 0.20, 0.25 mol from total mass of a mix CaCO_3_:TiO_2_. All chemicals were purchased from Sigma Aldrich (Sigma-Aldrich, Saint Louis, MO, USA), and chemical purity was 99.95%.

After suspension, the obtained mixtures were ground in a planetary mill for 1 h at a grinding speed of 400 rpm. The second step was the thermal annealing of the obtained mixtures for 5 h at an annealing temperature of 1300 °C, followed by cooling of the mixtures together in the furnace for 24 h. The choice of this method of synthesis, in spite of its being carrying out in several stages, was caused by the possibilities of its scaling and transfer to the industrial production of photo-catalysts in case producers and industrialists are interested in this method.

The morphological features of the synthesized lanthanum-doped ceramics were determined by scanning electron microscopy using a Hitachi TM 3030 microscope (Hitachi Ltd., Tokyo, Japan). To determine the elemental composition of the studied samples depending on the concentration of the dopant, the energy dispersive analysis method was used, implemented on a Hitachi TM 3030 microscope. The elemental composition was determined by mapping the samples, followed by determining the content of elements from 10 different areas to determine the average value.

The study of the phase transformation dynamics was carried out using the X-ray phase analysis method implemented on an X-ray diffractometer D8 Advance ECO (Bruker, Germany). The analysis was performed using the PDF-2 database (2016) and the Rietveld method used to refine the phase content and determine their contributions. The lattice parameters and the degree of structural ordering were refined using the Diffrac EVA v.4.2 software code.

The optical properties of the synthesized lanthanum-doped ceramics were studied by UV-Vis spectroscopy using a Jena Specord-250 spectrophotometer (Analytik Jena GmbH, Jena, Germany) in the wavelength range from 200 to 1000 nm, in steps of 1 nm.

The value of the band gap width (*E_g_*) was determined by plotting the Tauc plots and then calculating using Formula (1):(1)$$\alpha =A{\left(h\nu -E_{g}\right)}^{1/2},$$
where *A* is a constant and *hν* is the photon energy.

The refractive index (*n^optical^*) was calculated using Formula (2):(2)$$\frac{\left[{\left(n^{optical}\right)}^{2}-1\right]}{\left[{\left(n^{optical}\right)}^{2}+2\right]}=1-\sqrt{\frac{E_{g}}{20}}.$$

The values of optical transmission (*T^optical^*) and refraction loss (*R^loss^*), reflecting changes in optical transmission properties and reflection losses, were calculated using Formulas (3) and (4):(3)$$T^{optical}=\frac{2\left(n^{optical}\right)}{{\left(n^{optical}\right)}^{2}+1},$$
(4)$$R^{loss}={\left(\frac{\left(n^{optical}\right)-1}{\left(n^{optical}\right)+1}\right)}^{2}.$$

The value of the static dielectric permeability was determined using Formula (5):(5)$$\varepsilon^{static}={\left(n^{optical}\right)}^{2}.$$

The determination of the strength properties of ceramics depending on the dopant concentration was carried out using the indentation method. A Vickers pyramid was used as an indenter and measurements were carried out at a load of 100 N on the indenter. The determination of crack resistance was carried out using the method of single compression at a compression rate of 0.2 mm/min.

The efficiency of the synthesized ceramics as photocatalysts was evaluated using a model reaction of photocatalytic decomposition of the organic dye Rodamine B. The photocatalytic reactions were performed using a borosilicate glass reactor placed in a water bath in order to avoid overheating of the model solution and to exclude temperature factors that could affect the decomposition rate. UV exposure was simulated using a xenon lamp (2100 lm and 500 W) placed above the reactor at a distance of 5 cm. The initial dye concentrations were 20 mg/L and the test time interval was 300 min. Efficiency was measured by taking absorption spectra in the region of 400–700 nm, which is typical of the spectral lines of the compound Rodamine B. Efficiency was evaluated using Formula (6):(6)$$Degradation\;\;efficiency=(1-\frac{C_{i}}{C_{0}})\times 100\%,$$
where *C*_0_ and *C_i_* are absorption densities before and after the photocatalytic reaction.

The reaction rate constant, reflecting changes in the dye concentration during the reaction, was estimated according to the Ber-Lambert law (7):(7)$$kt=-\ln(\frac{C_{i}}{C_{0}}),$$
where *k* is the constant reaction rate, and *t* is the reaction time.

## 3. Results and Discussion

As shown earlier, in the case of mechanochemical synthesis of CaTiO_3_-type perovskite-like ceramics at annealing temperatures above 800 °C in the ceramics structure, the formation of TiO_2_-rutile phase impurity inclusions is observed, which presence is due to phase transformations and partial displacement of impurity from the main CaTiO_3_ phase with perovskite structure of orthorhombic type [29,30]. An increase in the annealing temperature leads to partial structural ordering of the TiO_2_ phase as well as an increase in the grain size characteristic of it. The presence of a rutile phase in CaTiO_3_ is observed only in the case of perovskite-like ceramics obtained by solid-phase synthesis.

When increasing the annealing temperature up to 1300 °C in the case of initial mixtures not containing lanthanum dopant, according to the analysis of X-ray diffractograms shown in Figure 1 we observe a number of diffraction reflexes in the region of 2θ = 27–28° which are typical for TiO_2_-rutile phase with orthorhombic type of lattice and P42/mnm(136) space group. The main reflexes on the diffractogram correspond to the CaTiO_3_ phase with perovskite structure and orthorhombic type of lattice, space group Pbnm(62). The content of TiO_2_-rutile phase in relation to the main phase of perovskite CaTiO_3_ is no more than 10%. Analyzing the width of diffraction reflexes with subsequent recalculation using the Scherrer formula to estimate the size of crystallites, it was found that the size of crystallites for the main CaTiO_3_ phase is more than 55 nm, while the size of crystallites for the impurity rutile phase is not more than 10–15 nm. This difference indicates that the formation of the rutile phase occurs by partial displacement of the TiO_2_ phase during the formation of the CaTiO_3_ perovskite structure, as well as by processes of phase transformations of the TiO_2_-anatase → TiO_2_–rutile type, which are typical at temperatures of 400–1000 °C. These transformations can be caused by a decrease in the number of oxygen vacancies, which leads to the formation of rutile-specific octahedrons or by compaction processes caused by structural ordering during thermal sintering of ceramics. The degree of structural ordering for the initial samples annealed at 1300 °C was more than 82%. Analysis of the structural parameters showed that the lattice parameters for the main CaTiO_3_ phase are a = 5.38582 Å, b = 5.42613 Å, c = 7.61653 Å, V = 223.0 Å^3^, for the TiO_2_ phase a = 4.62556 Å, c = 2.95844 Å, V = 63.30 Å^3^. The difference of crystal lattice parameters for both phases from reference values for the card values (COD-900-44-27 CaTiO_3_ a = 5.38 Å, b = 5.44 Å, c = 7.639 Å, COD-900-44-27 TiO_2_-Rutile, a = 4.603 Å, c = 2.966 Å) is caused by mechanochemical synthesis processes and the following thermal annealing of samples as well as by lattice deformation processes caused by them. Analyzing the character of changes in the parameters and their deviation from the reference values, we can arrive at the anisotropic character of the changes caused by tensile and compressive stresses occurring in the structure.

In the case of doped samples, the general dynamics of the changes in the diffraction reflexes indicate the processes of phase transformation with increasing dopant concentration, which is evidenced by the appearance of new diffraction reflexes, as well as the main shift observed in the original sample reflexes characteristic of the CaTiO_3_ phase. At lanthanum doping with concentration *x* = 0.05, the main changes in diffraction reflexes observed on the diffractogram correspond to the change of the CaTiO_3_ and TiO_2_ phases ratio, with the characteristic decrease of TiO_2_ phase content in ceramics composition. For these samples low-intensity diffraction reflexes are also observed in the region of 2θ = 28–33°, the presence of which can be associated with the formation of new phases, but their low intensity does not allow accurate identification of them as full-fledged phases.

When the dopant concentration was increased to *x* = 0.10, the low-intensity reflexes established earlier and unidentified became more pronounced, which allowed us to establish their correspondence with the La_2_TiO_5_ phase with the orthorhombic type of structure, as well as the Pnam(62) space group. The identification of this phase was made using the PDF-2 database, and the most characteristic card was COD-100-81-55. Meanwhile, the reflexes previously corresponding to the rutile phase in the case of dopant concentration *x* = 0.10 have undergone a shift and correspond to the La_2_TiO_5_ phase, which indicates the process of replacement of the TiO_2_ phase by La_2_TiO_5_ phase, for which the concentration in the analysis of diffraction reflex contributions was over 15%. 

For samples with dopant concentration *x* = 0.15 mol, no visible changes associated with phase transformations were observed, while the La_2_TiO_5_ phase contribution increased to 25%, indicating that with increasing dopant concentration a partial substitution of calcium or titanium with subsequent formation of the La_2_TiO_5_ phase by lanthanum occurs. The presence of this La_2_TiO_5_ phase characterized by a complex oxide structure characteristic of rutile phase substitution is usually observed at high annealing temperatures and is due to the high-temperature interaction of TiO_2_ and La_2_O_3_ oxides, the formation of which may be due to transient processes during mechanical agitation of La(NO_3_)_3_ or thermal annealing.

In the case of dopant concentration *x* = 0.20 mol the formation of the substitution phase of La_0.3_Ca_0.7_TiO_3_ type orthorhombic crystal structure with space group Pbnm(62), as well as the basic phase CaTiO_3_, is observed. Herewith, the reflexes characteristic of La_2_TiO_5_ complex oxide are not observed. The formation of the La_0.3_Ca_0.7_TiO_3_ phase is characteristic of the processes of replacement of calcium ions by lanthanum ions, with subsequent formation of a perovskite structure, as well as displacement of the impurity phases of complex oxides or rutile. The ratio of CaTiO_3_ and La_0.3_Ca_0.7_TiO_3_ phases is 51:49, which indicates a partial replacement of calcium by lanthanum and displacement of the CaTiO_3_ phase.

When increasing the dopant concentration up to *x* = 0.25 mol, the X-ray diffractograms show reflexes characteristic only of the La_0.3_Ca_0.7_TiO_3_ phase, which indicates that the processes of phase transformations are fully completed in the ceramics structure with the formation of the stable La_0.3_Ca_0.7_TiO_3_ phase as the final phase.

On the basis of the obtained data and the analysis, the dynamics of phase transformations depending on the concentration of lanthanum have been determined, which can be written down as follows: CaTiO_3_/TiO_2_ → CaTiO_3_/La_2_TiO_5_ → CaTiO_3_/La_0.3_Ca_0.7_TiO_3_ → La_0.3_Ca_0.7_TiO_3_. The dynamics of phase transformations and phase content depending on the dopant concentration are shown in Figure 2.

Table 1 presents data on the crystal lattice parameters of the studied samples depending on the dopant concentration in the structure.

Figure 3 shows the results of the morphological features of the synthesized ceramics depending on the dopant concentration.

The general appearance of particle morphology changes, depending on the dopant concentration, testifies to the processes of phase transformations, accompanied by the change in the shape and size of the grains of which the ceramics are composed. In case of underdoped ceramics, the morphological features of the synthesized structures consist of the formation of feather-like or columnar structures of asymmetric shape having transverse dimensions from several tens of nanometers to several hundreds of nanometers. The addition of lanthanum in a concentration of 0.05–0.10 mol leads to slight changes in the shape of the particles, as well as the formation of small grains of spherical shape whose dimensions do not exceed 10–20 nm. The formation of the La_0.3_Ca_0.7_TiO_3_ phase in the structure and its subsequent dominance leads to the formation of lamellar diamond-shaped particles which form dendritic structures. Therefore, we can see that the change in the phase composition is also accompanied by changes in the morphology and size of the grains.

Table 2 presents data on changes in the elemental composition of ceramics depending on the concentration of lanthanum dopant.

From the elemental composition data, it can be seen that in the case of underdoped ceramics the ratio of elements is close to the stoichiometric ratio characteristic of the CaTiO_3_ structure, with a small excess of oxygen, indicating the possible presence of oxygen vacancies or individual particles characteristic of the rutile phase in the structure. When adding lanthanum to the structure of ceramics, a significant increase in oxygen concentration is observed, which decreases as the dopant increases and the phase transformations follow. At the same time, the average value of the elements Ca and Ti is approximately the same for all synthesized ceramics, which indicates that the substitution of titanium or calcium by lanthanum has an equal probability, which is also confirmed by the results of phase analysis. According to the mapping results, the distribution of lanthanum in the structure is isotropic and uniform, which indicates that the doping occurs throughout the ceramics.

Figure 4 shows the results of measurements of the optical transmission spectra of the investigated ceramics depending on the concentration of lanthanum dopant. The general changes observed in the spectra indicate a change in the transmittance of the ceramics, as well as a shift of the fundamental absorption edge, the change of which indicates a change in the electronic density and the width of the forbidden zone. Such changes are related both to the processes of phase transformations caused by changes in the concentration of lanthanum dopant and morphological changes.

As can be seen from the data presented, with the addition of lanthanum with a concentration equal to 0.05 mole, there is an increase in the transmittance of ceramics, as well as a shift of the edge of fundamental absorption and an increase in the band gap width (see Data in Table 3). The appearance of the La_2_TiO_5_ phase in ceramics composition and the displacement of the TiO_2_ phase leads to a sharp decrease of transmittance of more than two times, which testifies to an increase in the absorbing (absorption) capacity of ceramics. At the same time, the formation of the La_2_TiO_5_ phase leads to a decrease in the value of the band gap width to less than 2 eV, which indicates a strong change in the electronic density of the ceramics. A decrease in transmittance and change of electronic density can also be caused by a decrease in concentration of the elements in ceramics composition, in particular, a decrease of oxygen concentration, which can reduce the number of oxygen vacancies in the ceramics’ structure, as well as form new absorbing centers. Further increase of the lanthanum dopant concentration in the ceramics structure leads to decrease in the ceramics transmittance and of the band gap width; however, formation of the La_0.3_Ca_0.7_TiO_3_ phase in the structure leads to an insignificant increase of transmittance, but in the case of this phase dominance in the structure, formation of a widely induced absorption band in the region of 400–650 nm, typical of the visible light region, is observed.

On the basis of the obtained transmission spectra the basic optical characteristics were calculated, the data of which are presented in Table 3. Formulas (1)–(5) were used for calculations.

As can be seen from the data presented, a change in the phase composition leads to a decrease in the band gap width, the change in which leads to an increase in the refractive index and static permittivity, which indicates an increase in the absorption capacity of ceramics, as well as a change in the electronic density and charge distribution in the structure. The change in the *R^loss^* value indicates that the change in the phase composition of the ceramics leads to a decrease in losses associated with the reflection and polarization of light during its interaction with the ceramics. In turn, the presence of wide absorption bands for single-phase La_0.3_Ca_0.7_TiO_3_ samples can have a significant impact on the photocatalytic activity of ceramics if they are used as a basis for photocatalysts.

One of the key parameters that characterize the resistance of ceramics to external influences is their mechanical hardness and crack resistance. These parameters characterize not only the resistance of ceramics to various mechanical effects that may arise during their operation, but also determine the permissible limits at which these materials can be operated.

To evaluate the strength characteristics, the synthesized ceramics were pressed into disks 10 mm in diameter and 2 mm thick, which were further tested to determine the hardness and fracture resistance under single compression of the samples.

Figure 5 shows the dependence of changes in hardness and fracture resistance of ceramics with variation of dopant components. The general trends of these changes testify to the positive effect of ceramics hardening depending on the phase composition, changing with dopant concentration variation.

As can be seen from the data presented in the presence of TiO_2_-type impurity inclusions in the structure, as well as increased oxygen content, the strength properties are quite low, which indicates low strength and resistance to external influences. Doping with lanthanum leads to an insignificant increase in hardness and cracking resistance as well as strengthening of ceramics. Displacement of TiO_2_ impurity inclusions from the structure and formation of La_2_TiO_5_ phase leads to more than 20–25% hardening and an increase in crack resistance by 30–40%. The increase in the contribution of the La_2_TiO_5_ phase in the ceramics structure also leads to an increase in structural ordering, which also affects the strength properties of the ceramics. Formation of La_0.3_Ca_0.7_TiO_3_ phase in the ceramics structure according to the data changes in strength properties leads to a sharp increase in strength and cracking resistance that can be associated with both hardening due to displacement of impurity phases and reduction of oxygen concentration in the ceramics structure, and change in the morphological features and dislocation density of ceramics. The change in dislocation density is associated primarily with a decrease in the size of crystallites as well as their geometric shapes, which leads to the dislocation hardening effect characteristic of nanostructured or nanoscale materials. Such a strongly pronounced dependence of changes in the strength properties on the phase composition of ceramics, which are caused by variations in the dopant concentration, indicates that the main contribution to the effects of softening is made by impurity inclusions, a decrease in the concentration of which determines the hardening effect and an increase in resistance to external influences.

Figure 6 shows the kinetic dependences of changes in the concentration of Rodamine B decomposition by synthesized ceramics of different phase composition. These dependences were plotted based on the data on the change in the optical density of the absorption spectra and further calculation of the changes in the C/C_0_ concentration ratio as a function of the photocatalytic reaction time.

As can be seen from the presented data, the time dependences of the kinetics of changes in the C/C_0_ value have two characteristic sections characterized by a different slope of the curve, which indicates a different rate of the photocatalytic reaction. Such dependences are observed for samples with dopant concentrations of 0.05–0.15 mol. For these samples, we can see that, in the time interval of 0–180 min, the degree of decomposition changes sharply enough with time, indicating that the reactions proceed quite intensively. When the reaction time reaches 210 min, a characteristic plateau is observed for which the change in ∆C/C_0_ is no more than 3–7%. This behavior of the C/C_0_ decrease can be explained by several factors, which will be investigated in detail in further future studies.

One of the factors influencing such behavior of photocatalytic reactions can be the processes of partial absorption of dye decay product residues on the surface of photocatalysts, which reduces their efficiency. In this case also, a partial amorphization of the ceramics surface can occur, which can result in changes in their optical properties which, as well as the stability of the structural properties, are significantly influenced by the existing unstable impurities in the ceramics’ composition. In turn, the influence of impurities on the photocatalytic activity is evidenced by the results of the kinetic curves for the ceramics in which no impurity inclusions were observed.

In the case of ceramics with composition La_0.3_Ca_0.7_TiO_3,_ decomposition efficiency is constant, and at achievement of 300 min of course of reaction, the degree of decomposition is more than 95–97%. According to the data shown in Figure 6b, the decomposition efficiency as well as the reaction rate has a strong dependence on the phase composition of ceramics and the presence of impurity inclusions. The value of the reaction rate calculated using Formula (7), depending on the ceramics composition, changes by two–three times when the impurity phases are displaced from the ceramics’ composition.

Figure 7 shows the results of changes in the values of photocatalytic decomposition efficiency and mineralization of organic dye decomposition products after 300 min of photocatalytic reactions for all tested lanthanum-doped ceramics.

As can be seen from the data presented, the change in phase composition has a significant impact on the efficiency of photocatalytic reactions. Hence, displacement of the TiO_2_ phase from the composition leads to an increase in the decomposition efficiency from 60% to 75–80% and an increase in the degree of mineralization from 37% to 45–50% in the case of La_2_TiO_5_ phase formation. Such a strong difference in decomposition efficiency and degree of mineralization is due to the fact that, when dyes decompose, some of them are able to be absorbed on the surface of photocatalysts, which leads to the fact that part of the dye is not decomposed into harmless components, and the optical density of the solution is reduced. In this case, the replacement of the TiO_2_ phase by the La_2_TiO_5_ phase leads to an increase in the decomposition efficiency as well as the decomposition rate. The formation of La_0.3_Ca_0.7_TiO_3_ phase in the structure leads to an increase in the decomposition efficiency of up to 83–97% and in the degree of mineralization of up to 60–80%, which in the case of the decomposition of Rodamine B, which is one of the most resistant to degradation dye, is a good enough result for photocatalytic reactions. One of the factors influencing the decomposition efficiency and increasing the reaction rate is a change in the optical properties and the band gap width, decrease in which significantly accelerates the formation of photoelectrons required for the flow of photocatalytic reactions. In the case of changes in the phase composition of the ceramics, according to the optical properties data, the displacement of impurity phases and phase transformation of the CaTiO_3_/La_2_TiO_5_ → CaTiO_3_/La_0.3_Ca_0.7_TiO_3_ → La_0.3_Ca_0.7_TiO_3_ type leads to a decrease in the band gap width from 2.35–2.43 eV to 1.62–1.71 eV. Such a change leads to an increase in efficiency from 60% to 95–97%, which indicates a high efficiency of organic dye decomposition.

It is also worth noting that tests of synthesized ceramics to preserve decomposition efficiency and degree of mineralization during several cycling tests showed preservation of up to 90% efficiency after five cycles for ceramics with dopant concentration of 0.20–0.25 mol, while the efficiency of ceramics containing impurity TiO_2_ inclusions after five cycles was less than 50% of the initial efficiency value after one cycle (See results on Figure 8).

Figure 9 shows the results of ceramic surface morphology after 5 test cycles.

As can be seen from the presented data, in the case of undoped ceramics, a strong amorphization of the ceramic surface is observed due to the formation of microcracks, as well as a large amount of absorbed substances. At the same time, for samples containing lanthanum with a concentration of more than 0.10 mol, the presence of microcracks in the structure was not established, which indicates a rather high resistance of ceramics to degradation processes associated with photocatalytic reactions. An insignificant amount of absorbed inclusions is also observed on the surface of samples with a lanthanum concentration of more than 0.15 mol.

## 4. Conclusions

The paper considers the method of preparation of CaTiO_3_ titanates doped with lanthanum using the method of mechanochemical synthesis and subsequent thermal annealing, as well as their application, as a basis for photocatalysts for the decomposition of the organic dye Rodamine B. In the course of these studies, the dynamics of changes in morphological features and phase composition from the concentration of lanthanum dopant were established. Using the method of X-ray phase analysis, the dynamics of phase transformations depending on lanthanum concentration were established: CaTiO_3_/TiO_2_ → CaTiO_3_/La_2_TiO_5_ → CaTiO_3_/La0.3Ca_0.7_TiO_3_ → La_0.3_Ca_0.7_TiO_3_. In the course of determining the optical properties of the ceramics it was found that the change in phase composition due to the displacement of impurity inclusions leads to a decrease in the bandgap width, as well as changes in the electronic density and charge distribution in the structure. The results of photocatalytic reactions showed that ceramics with no impurity inclusions have the highest decomposition efficiency of the organic dye (Rodamine B).

## Figures and Tables

**Figure 1 nanomaterials-12-02241-f001:**
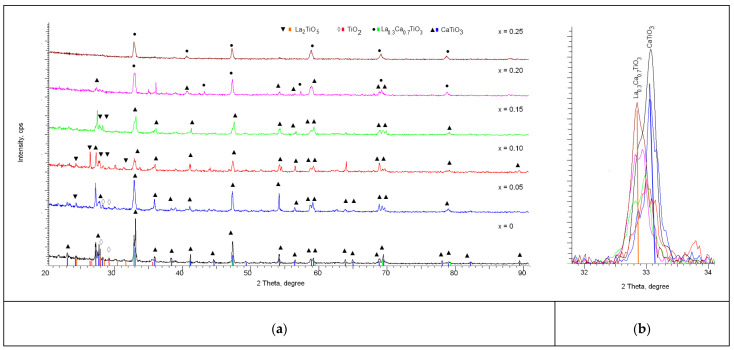
(**a**) X-ray diffractograms of synthesized samples of CaTiO_3_ perovskite-like ceramics doped with lanthanum (La); (**b**) Detailed change of the main reflex.

**Figure 2 nanomaterials-12-02241-f002:**
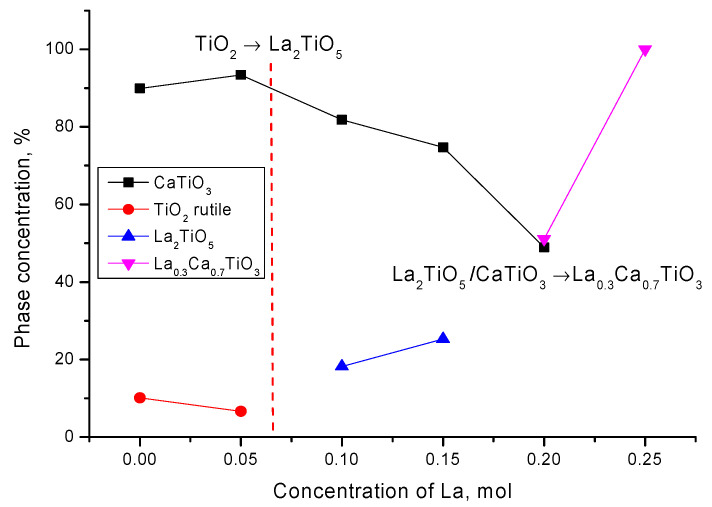
Scheme of phase relations of synthesized samples of CaTiO_3_ perovskite-like ceramics doped with lanthanum (La).

**Figure 3 nanomaterials-12-02241-f003:**
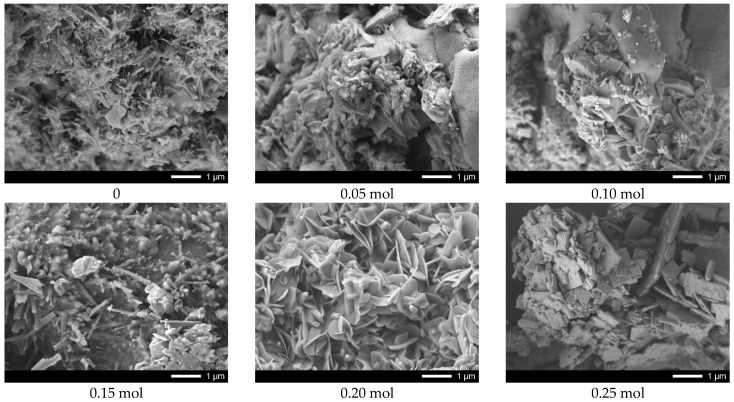
SEM images of synthesized CaTiO_3_ ceramics doped with lanthanum (La).

**Figure 4 nanomaterials-12-02241-f004:**
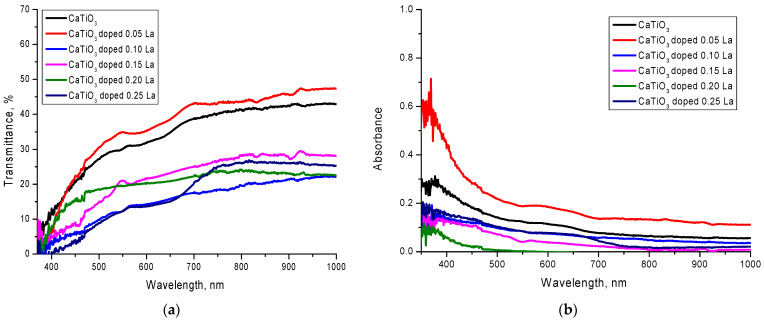
UV-Vis optical spectra of the investigated lanthanum-doped CaTiO_3_ ceramics: (**a**) Transmittance; (**b**) Absorbance.

**Figure 5 nanomaterials-12-02241-f005:**
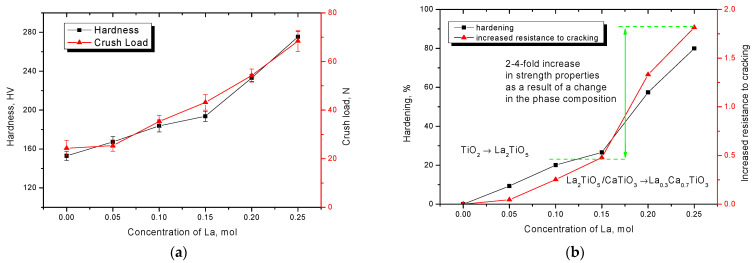
(**a**) Results of changes in hardness and crack resistance of ceramics; (**b**) Diagram of hardening of ceramics depending on phase composition.

**Figure 6 nanomaterials-12-02241-f006:**
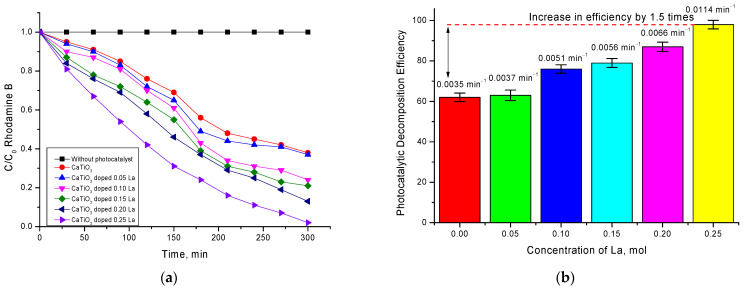
(**a**) Diagrams of Rodamine B concentration dependence on photocatalytic decomposition time; (**b**) Comparative diagram of photocatalytic decomposition efficiency of Rodamine B and reaction rate.

**Figure 7 nanomaterials-12-02241-f007:**
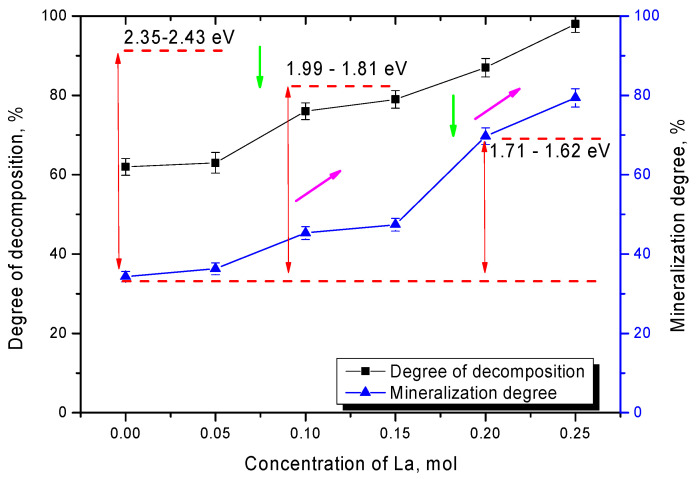
Dynamics of the dependence of the values of decomposition efficiency and mineralization of decomposition products on type of ceramics, as well as changes in the band gap width of ceramics (green arrows indicate changes in the band gap width; crimson arrows reflect the dynamics of increasing the photocatalytic decomposition efficiency).

**Figure 8 nanomaterials-12-02241-f008:**
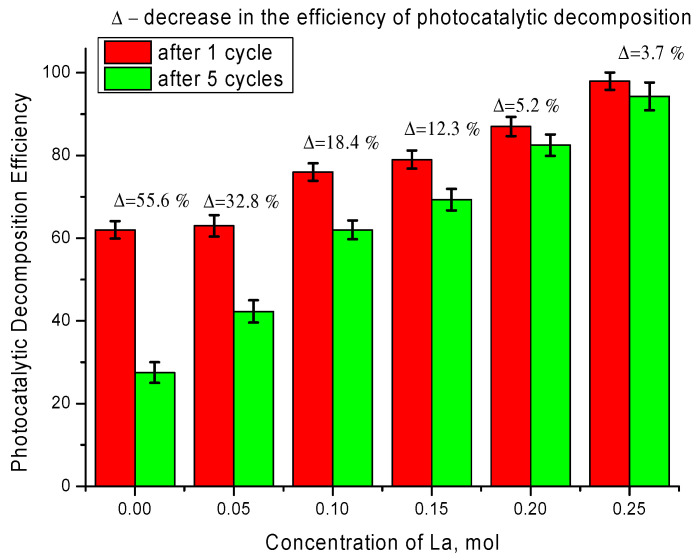
Results of evaluating the efficiency of photocatalytic activity after 1 and 5 cycles of successive reactions.

**Figure 9 nanomaterials-12-02241-f009:**
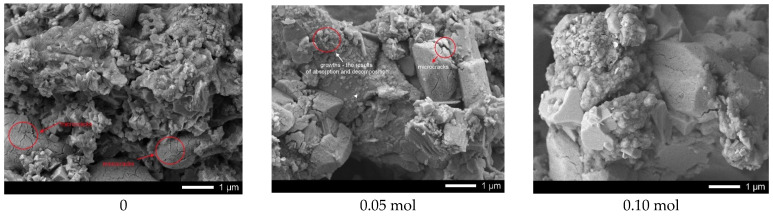
Results of ceramic surface morphology after 5 test cycles.

**Table 1 nanomaterials-12-02241-t001:** Data of crystal lattice parameters.

Sample	Concentration of La, mol					
	0	0.05	0.10	0.15	0.20	0.25
CaTiO_3_—orthorhombic	a = 5.39582 ± 0.0013 Å, b = 5.4261 ± 0.0017 Å, c = 7.6165 ± 0.0019 Å	a = 5.4109 ± 0.0017 Å, b = 5.4370 ± 0.0019 Å, c = 7.6257 ± 0.0023 Å	a = 5.4299 ± 0.0021 Å, b = 5.4497 ± 0.0025 Å, c = 7.6766 ± 0.0016 Å	a = 5.4022 ± 0.0015 Å, b = 5.4380 ± 0.0016 Å, c = 7.6630 ± 0.0018 Å	a = 5.3967 ± 0.0011 Å, b = 5.4318 ± 0.0013 Å, c = 7.6346 ± 0.0016 Å	-
TiO_2_—rutile, tetragonal	a = 4.6256 ± 0.0011 Å, c = 2.9584 ± 0.0016 Å	a = 4.6318 ± 0.0011 Å, c = 2.9608 ± 0.0021 Å	-	-	-	-
La_2_TiO_5_—orthorhombic	-	-	a = 11.0109 ± 0.0018 Å, b = 11.4257 ± 0.0013 Å, c = 3.9446 ± 0.0023 Å	a = 11.0389 ± 0.0018 Å, b = 11.3921 ± 0.0014 Å, c = 3.9501 ± 0.0011 Å	-	-
La_0.3_Ca_0.7_TiO_3_—orthorhombic	-	-	-	-	A = 5.4189 ± 0.0017 Å, b = 5.4566 ± 0.0011 Å, c = 7.7046 ± 0.0023 Å	a = 5.4029 ± 0.0022 Å, b = 5.4491 ± 0.0015 Å, c = 7.6910 ± 0.0013 Å
Density, g/cm^3^	4.006	3.996	4.256	4.364	4.213	4.264

**Table 2 nanomaterials-12-02241-t002:** Data of elemental analysis.

Sample	Concentration of La, mol					
	0	0.05	0.10	0.15	0.20	0.25
Ca, at.%	16.62 ± 0.81	10.15 ± 0.42	10.25 ± 0.52	11.13 ± 0.51	11.75 ± 0.61	11.71 ± 0.53
Ti, at.%	13.68 ± 0.82	8.70 ± 0.41	9.87 ± 0.56	11.35 ± 0.45	11.23 ± 0.62	16.43 ± 0.82
O, at.%	69.68 ± 6.11	80.94 ± 6.31	79.60 ± 7.11	76.90 ± 4.21	75.63 ± 6.42	70.00 ± 5.32
La, at.%	-	0.21 ± 0.11	0.28 ± 0.11	0.62 ± 0.11	1.40 ± 0.21	1.86 ± 0.32

**Table 3 nanomaterials-12-02241-t003:** Data of optical characteristics.

Sample	Concentration of La, mol					
	0	0.05	0.10	0.15	0.20	0.25
*E_g_*	2.352	2.436	1.994	1.809	1.711	1.623
*n^optical^*	2.597	2.568	2.738	2.824	2.873	2.921
*T^optical^*	0.197	0.193	0.216	0.227	0.233	0.240
*R^loss^*	0.671	0.676	0.644	0.629	0.620	0.612
*ε^static^*	6.741	6.594	7.496	7.974	8.254	8.532

## Data Availability

Data presented in this article are available at request from the corresponding author.

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
