# Peer review of "Synthesis, Properties and Photocatalytic Activity of CaTiO3-Based Ceramics Doped with Lanthanum"

_nanomaterials, 2022, doi:10.3390/nano12132241_

Round 1
Reviewer 1 Report
Dear Authors, thank you for your manuscript, submitted to "Nanomaterials". After the reading the manuscript, I propose to attract your attention to the following questions and comments:
1. From the Introduction Part it is not clear, why is the necessity to modify the titanates as photocatalysts? Why was La chosen as a doping element? The mentioned references [25-28] could not answer this question.
2. Experimental Part. Please, add the analytical grade of the initial reagents.
3. Please, testify the content of La dopant - in Experimental Part mass. % was specified, through a whole text - mol, in Table 2 - at. %. It is uncomfortable to evaluate La doping as a result.
4. Pbnm is not "spatial syngony". It is space group. Syngony is orthorombic.
5. Please, approve "partial ordering of the TiO2 phase". What kind of ordering is being talking about?
6. Sentense "The content of this phase in the relation to the main phase.. " is out of place. It must be located above - there is "this phase", TiO2.
7. What structure does the initial reagent TiO2 have?
8. Could you please add any symbols for different phases to XRD patterns at Fig. 1?
9. Fig. 2 is not the phase diagram. It is the scheme of phase relations.
10. Please, add the errors to table 1.
11. What was method used for elemental analysis?
12. Could you please illustrate the Rietveld refinement of the XRD data for single-phase sample La0.3Ca0.7TiO3?
13. What was the density of ceramics?
14. The resulting Part requires the discussion and comparison with existing literature data, for example, it is wishable to compare data of Table 6 and Fig. 5 with literature data.
15. Is it possible to investigate the surface of photocatalysts after dyes decompose for approving decomposition efficiency and definition of mineralization degree?
Author Response
Please see the attachement.

Reviewer 2 Report
In this manuscript, CaTiO3 titanates doped with lanthanum was prepared using the method of mechanochemical synthesis and subsequent thermal annealing, and the performance for decomposition of the organic dye Rhodamine B was studied. However, this paper renders quite routine works which lead to the lacking of novelty. Some explanations need to be further discussed and some details should be corrected.
1. In Figure 1, it is better to use relative intensity when comparing the XRD diffraction peaks of different samples. The expression in Figure 1 may mislead readers.
2. The standard XRD pattern of the reference material needs to be provided.
3. The actual color change of samples and UV-Vis diffuse reflectance spectrum should be provided to visually represent the band edge change.
4. The stability of the photocatalytic degradation of organic dye Rhodamine B should be provided.
5. The authors claim that electron density and charge distribution change in the ceramic structure, and the supporting evidence should be given. It is recommended to include time-resolved PL (TRPL) measurements, as it can provide intrinsic insights into the carrier lifetime and the material's electron distribution.
Round 2
Reviewer 1 Report
Dear Authors, thank you for your attention to my remarks!
Reviewer 2 Report
The author has raised most of the questions, I recommend the acceptance of this paper.